# Thyroid Carcinoma Showing Thymus-like Differentiation (CASTLE): A Case Report

**DOI:** 10.3390/life12091314

**Published:** 2022-08-26

**Authors:** Mihaela Stanciu, Ruxandra Paula Ristea, Mihaela Popescu, Corina Maria Vasile, Florina Ligia Popa

**Affiliations:** 1Department of Endocrinology, Faculty of Medicine, Lucian Blaga University of Sibiu, 550169 Sibiu, Romania; 2Department of Endocrinology, County Clinical Emergency Hospital of Sibiu, 550245 Sibiu, Romania; 3Department of Endocrinology, University of Medicine and Pharmacy of Craiova, 200349 Craiova, Romania; 4Department of Pediatric Cardiology, “Marie Curie” Emergency Children’s Hospital, 041451 Bucharest, Romania; 5Department of Physical Medicine and Rehabilitation, Faculty of Medicine, Lucian Blaga University of Sibiu, 550169 Sibiu, Romania

**Keywords:** thyroid carcinoma, immunohistochemistry, thyroidectomy, thymus-like differentiation

## Abstract

Background and Objectives: Carcinoma showing thymus-like differentiation (CASTLE) is a low-grade thyroid carcinoma, with an indolent clinical course and usually a favorable prognosis. The clinical and imagistic features are not specific for CASTLE but similar to other malignant lesions of the thyroid. Definite diagnosis is based on an immunohistochemical examination, as this carcinoma shows positive CD5 immunoreactivity when compared to other aggressive thyroid carcinomas. Case presentation: The main focus of this study is to outline a rare case of CASTLE compressing the trachea in a 50-year-old female patient who was initially diagnosed with undifferentiated thyroid carcinoma, for which she underwent unsuccessful surgery, as well as postoperative radiotherapy and chemotherapy. After receiving a second medical opinion, the patient underwent a challenging radical resection consisting in total thyroidectomy and central neck dissection, with no local recurrence after 6 months and 2 years of follow-up and negative metastatic follow-up. The correct diagnosis has been established based on pathological and immunohistochemical examinations. Conclusions: In summary, the diagnosis of CASTLE is difficult and requires an experienced histological analysis and CD5 immunoreactivity. Lack of metastasis, complete removal of the tumor, and a low degree of tumor infiltration into nearby structures are all associated with better long-term survival.

## 1. Introduction

Carcinoma showing thymus-like differentiation (CASTLE) is a rare, low-grade thyroid malignancy. The incidence of CASTLE reported worldwide among thyroid malignancies is estimated to be less than 0.1% [1,2,3,4,5]. Both genders are similarly affected, though there is a slight female predominance, with an F:M ratio of 1.3:1 [6,7]. It is most common during the fourth and fifth decades of life, with the average age at onset occurring around 48.5 years [6,7,8].

In 2004, the World Health Organization (WHO) classified CASTLE as a clinically and pathologically independent thyroid tumor entity, after it was first described as an “intrathyroidal epithelial thymoma” in 1985 by Miyauchi et al. [9,10]. Histologically, CASTLE is very similar to thymic tissue. It is thought to have its origin either from ectopic thymus tissue or from remnants related to thymus development in or adjacent to the thyroid. This hypothesis is based on the fact that the tumor usually arises at the lower pole of the thyroid and exhibits several features of thymic differentiation, such as lobulation on cut surfaces; an expansive growth pattern; thick, fibrous bands dividing tumor cell nests; many lymphocytes; perivascular spaces with lymphocytes; sparse or rare mitoses; and oval, vesicular nuclei, well-defined nucleoli, and pale cytoplasm. This tumor is also lacking foci of papillary, follicular, medullary, or anaplastic carcinoma [11].

The diagnosis is challenging as carcinomas showing thymus-like differentiation have similar clinical characteristics and imagistic features compared to other aggressive thyroid carcinomas. The conclusive diagnosis requires pathological examination and positive cluster of differentiation 5 (CD5) immunoreactivity [12]. In most cases, the immunohistochemical test shows CD5 positive staining and negative staining for thyroid gland markers such as thyroglobulin and calcitonin. Molecular analysis reveals that most thymic tumors are p63-positive, whereas cystic carcinoma and poorly differentiated thyroid forms are negative [3]. Therefore, CD5 is used as a marker of thymic origin. Ito et al. [11] reported a sensitivity and specificity of 82% and 100%, respectively, for CD5-based detection of CASTLE. Although its negative value does not completely exclude the possibility of a CASTLE diagnosis, CD5 can definitely support the CASTLE assessment.

An experienced histologist requests the performance of CD5 when the histopathological examination cannot clearly establish differentiated cancer (e.g., papillary, follicular) or medullary cancer and when the usual immunohistochemical panel shows negative results for calcitonin, P63, synaptophysin, thyroglobulin, and chromogranin.

Correct diagnosis is essential as CASTLE has a positive prognosis compared to other thyroid carcinomas, and complete resection of the tumor followed by postoperative radiotherapy is considered effective in improving long-term survival [6].

We outline the case of a patient diagnosed with CASTLE who suffered tracheal compression after being initially diagnosed with undifferentiated thyroid carcinoma, for which she underwent unsuccessful surgery and both postoperative radiotherapy and chemotherapy.

## 2. Case Report

After being initially diagnosed with undifferentiated thyroid carcinoma in a different department, a 50-year-old female patient was referred to our Endocrinology Clinic for further investigation of an 18-month history of persistent dyspnea, dry cough, loss of weight, lack of appetite, and fatigue, associated with dysphagia and dysphonia. Family history of thyroid pathology was denied. Before the presentation, the patient noted a painless, growing cervical mass and accused dry cough, dysphonia, dysphagia, dyspnea, and fatigue.

Her medical history begins with a cervical ultrasound (US) that revealed a hypoechogenic, finely inhomogeneous right thyroid lobe, measuring 18.5/33/42.5 mm (AP/T/L) and the left thyroid lobe entirely occupied by a nodular lesion, with a hypoechoic and inhomogeneous echotexture, measuring 41.5/35/31 mm (Figure 1). The serum concentrations of thyroid-stimulating hormone (TSH), free-thyroxine-4 (FT4), thyroglobulin (TG), calcitonin, and antithyroid peroxidase antibodies (ATPO) were normal.

Total thyroidectomy was therefore recommended. Surgical resection was intended, but on exploration a 4 cm thyroid mass with macroscopically tumoral appearance, extensively invasive in the trachea, hyoid muscles, and left internal jugular vein was found. Surgical intervention was limited to biopsy. Pathological examination confirmed the diagnosis of undifferentiated thyroid carcinoma with squamous differentiation. Subsequent immunohistochemical staining showed positivity for tumor protein (p63), cytokeratin 19 (CK19), cytokeratin AE1/AE3 (CKAE1/AE3), and negativity for thyroid transcription factor-1 (TTF1) (Figure 2).

There was no cluster of differentiation (CD5). Computerized tomography (CT) of the neck and chest revealed the related lesion as a hypodense mass with irregular margins, completely invading the left thyroid lobe, compressing the trachea, and in close contact with the esophagus, measuring 3/3.5/3.7 cm (Figure 3).

The patient was referred to an oncology service with a poor prognosis and underwent both radiotherapy and chemotherapy. However, our patient considered it appropriate to request a second opinion and presented to our service. Moreover, the management of the case included a reassessment of the diagnosis and a surgical re-evaluation. A CT scan of the neck and chest was performed. The results revealed the same pattern, with the lesion occupying the lower left thyroid lobe, extending into the trachea, esophagus, and left inferior jugular vein, compressing the lumen of the trachea. The patient underwent a challenging radical surgery in which total thyroidectomy and cervical lymphadenectomy were performed. The procedure was quite radical with the removal of the tumor from the compression zone of the esophagus, with minimal reconstruction of the esophagus while an intragastric PEG probe was placed to feed the patient. This probe was left for 1 month and then removed with the restoration of digestive transit. Macroscopically, the lesion had irregular margins and was adherent to adjacent structures, but still without invasion into the trachea. Complete resection of the tumor was successfully performed.

After surgery, adjuvant radiotherapy was not required, and replacement therapy with Levothyroxine 75 mcg/day was initiated. No acute complications were reported. Metastatic follow-up was negative and there was no evidence of loco-regional recurrence after 6 months and 2 years of follow-up. Large areas of fibrosis, large, round, vesicular nuclei with small nucleoli, low cytoplasm, and peritumoral lymphoplasmacytic infiltration were revealed upon histopathological examination of the sample. CD5 immunostaining was positive (Figure 4).

## 3. Discussion

Carcinoma with thymus-like differentiation (CASTLE) is considered a rare malignant neoplasm entity, typically located at the level of the thyroid gland with occasional invasion of surrounding structures [13].

In 1991, Chan and Rosai provided a classification based on four groups of thymic differentiating neck tumors: carcinoma exhibiting thymus-like differentiation (CASTLE), spindle-shaped epithelial tumor with thymus-like differentiation (SETTLE), ectopic hematomatous thymoma, and ectopic cervical thymoma. The last two are considered benign [14,15,16].

Based on the literature, CASTLE has an average age at diagnosis of 50 years, with a female-to-male ratio of 1.3:1 [6,7,8,11]. Spindle-shaped epithelial tumor with thymus-like differentiation (SETTLE) is an unusual thyroid tumor common in the pediatric and young adult age group [12]. It predominantly occurs in young people, especially children, adolescents, and young adults, but may be found in the middle age as it has been seen in patients aged 4–59 years [13].

Because the clinical features are not specific, the most common symptoms are hoarseness, dysphagia, and a palpable, hard mass located around the throat with reduced mobility. Metastasis to regional lymph nodes and invasion into adjacent structures may already be present at the time of diagnosis. Incidence of tracheal invasion is between 24–38%. CASTLE tends to be located primarily in the lower part of the cervical trachea, as it appears to originate from the lower poles of the thyroid gland [11]. This distribution causes respiratory symptoms rather frequently, including bloody sputum and dyspnea [11,17].

These clinical findings are also present in other types of aggressive thyroid carcinomas (e.g., undifferentiated carcinoma, squamous cell carcinoma), underlying the difficulty of diagnosing CASTLE. However, despite the metastatic potential, it usually has a positive prognosis, with a survival rate of 90% and 82% for 5 and 10 years, respectively [7,18].

Thyroid carcinoma diagnosis is based on cervical ultrasonography (US), computed tomography (CT) of the neck, fine-needle aspiration cytology (FNAC), and thyroid function investigation. However, thyroid function is usually unaltered [6]. Diagnostic imaging techniques for CASTLE are not specific for the diagnosis. Ultrasound (US) examination of the thyroid may reveal a solid lobulated hypoechoic mass with moderate vascularity that does not show calcification. The mass usually originates from the lower part of the thyroid and only slow enlargement of such a lesion can raise suspicion for CASTLE [19]. Yamamoto et al. reported ultrasound features such as heterogeneously solid tumors lacking cystic components or calcification, and the central part of the tumor was reported to be slightly hyperechoic when compared to the peripheral part of the tumor [20].

In our case, US cervical ultrasound shows non-specific features, the thyroid mass is usually seen as hypoechoic, with a heterogeneous pattern, irregular margins, moderate vascularity, small cystic components, and no calcifications, which was not conclusive for CASTLE. Computed tomography of the neck usually reveals a well-defined mass with irregular margins and soft density [14]. On non-contrast computed tomography, tumors are poorly defined and nodular with constant density. They have attenuation comparable to that of adjacent muscle, but do not show calcification. Tumor cysts are unusual. On contrast-enhanced CT scans, tumors usually show only mild heterogeneous enhancement [21]. According to the above, the results of our patient’s cervical ultrasound and CT scans were described in consistency with the literature.

The accurate diagnosis of CASTLE is based on pathological and immunohistochemical examinations. Both CASTLE and thymic carcinoma show immunoreactivity for CD5, p63, CD117, and CK19, and are negative for thyroid tissue markers: TG, TTF-1, and calcitonin [18]. Most thymomas and other thyroid malignant tumors present negative CD5 immunoreactivity [22,23]. It has been reported that high molecular weight keratin (HMWCK), carcinoembryonic antigen (CEA), and p63 expression in CASTLE are evidence of thymic origin and can be used to differentiate thyroid CASTLE from other thyroid neoplasms [20]. Ito et al. reported a sensitivity and specificity of 82% and 100%, respectively, for CD5 positive for CASTLE diagnosis [7]. In a study performed by Reimann et al. they showed that CD5, HMWCK, CEA and p63 expression in CASTLE were valuable markers to confirm the diagnosis [24]. Histologically, the tumor presents a lobulated pattern; distinct borders from the surrounding thyroid tissue; peritumoral lymphoplasmacytic infiltration; dense fibrous septa, which divide the tumor nests; and medium or large vesicular nuclei, mostly oval, with prominent nucleoli [25].

Management of CASTLE is still not definitive due to its rarity. There are no current standard treatment guidelines, but surgery is usually considered the first option. It should be distinguished from other types of aggressive thyroid tumors, as CASTLE shows a favorable prognosis after radical surgery and postoperative radiation to minimize the locoregional recurrence rate. In the literature, recurrence rates ranging from 14% to 21% have been reported. Local invasion involving the trachea and esophagus did not result in a substantial difference in overall prognosis [7,26]. In a review of 117 cases diagnosed as thymic tumors, Curran et al. demonstrated that postoperative radiotherapy was able to provide better local management than radical surgery at its own. The presence of metastases and extension to adjacent structures (e.g., larynx, trachea, esophagus, vascular structures) are considered important prognostic indicators [27]. First therapeutic option is radical surgical resection consisting of total thyroidectomy and neck dissection. Tumor invasion can make the procedure even more difficult [6].

There is evidence in the literature that postoperative radiotherapy may prevent loco-regional recurrence [5,10,28,29]. Furthermore, several authors have suggested that postoperative radiotherapy should be considered for patients with either positive or suspected lymph node involvement. According to the study performed by Roka and Piacentini, it seems that surgery is enough for patients without lymph node metastases, as not one of the patients in their series suffered from recurrence [5,30]. In a study performed by Tsutsui H et al. [3], 2 patients who had no lymph node metastases and did not receive postoperative radiotherapy were follow-up for 5 and 10 years, respectively, with no relapse. Sun et al. showed that there is rarely recurrence after removal of lymph nodes at initial surgery with negative pathological results [31]. For patients who suffer from recurrence after initial treatment, surgery and radiotherapy still play an important role rather than chemotherapy. Some reports have shown lower rates of local recurrence in a group of patients who had local invasive tumors and cervical lymph node metastases when being treated with radiotherapy [6]. Our patient has not required postoperative adjuvant radiotherapy.

## 4. Conclusions

An essential role in the diagnosis of CASTLE is achieved by pathological and immunohistochemical examinations, but the gold standard remains positive CD5 immunoreactivity.

To improve prognosis and prevent loco-regional recurrence requires complete resection of these tumors and postoperative radiotherapy for a positive outcome.

This case is particular because it demonstrates the value of establishing an accurate diagnosis to achieve optimal management in order to prevent unnecessary procedures.

## Figures and Tables

**Figure 1 life-12-01314-f001:**
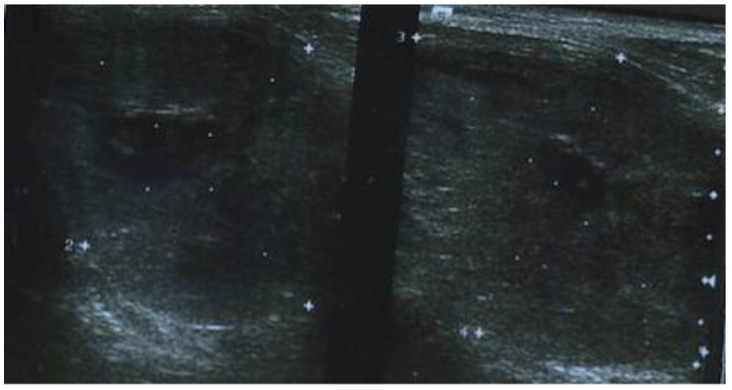
Cervical ultrasonography showing a hypoechogenic, finely inhomogeneous right thyroid lobe and the left thyroid lobe entirely occupied by a nodular lesion, with a hypoechoic and inhomogeneous echotexture.

**Figure 2 life-12-01314-f002:**
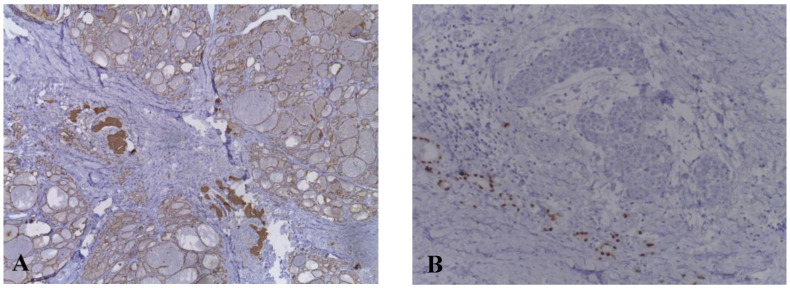
Immunohistochemical examination of carcinoma showing thymus-like differentiation. (**A**) Tumor cells are positive for CKAE1/AE3. (**B**) Tumor cells are negative for TTF1.

**Figure 3 life-12-01314-f003:**
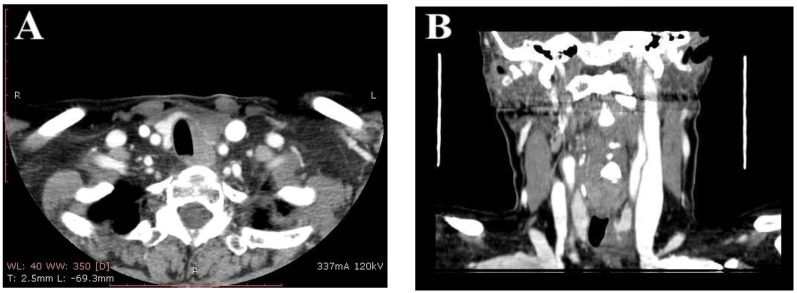
(**A**) Contrast enhanced neck computed tomography scan showing a hypodense mass located in the thyroid gland, with irregular borders, that compresses the trachea. (**B**) Coronal neck computed tomography scan showing corresponding thyroid mass.

**Figure 4 life-12-01314-f004:**
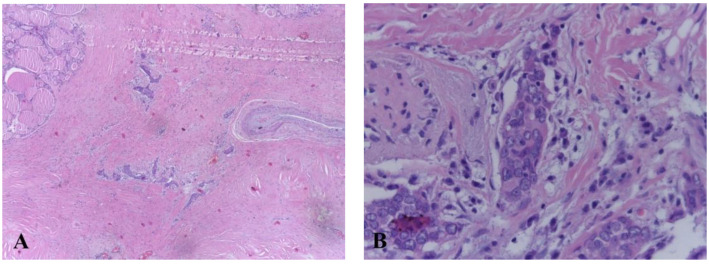
(**A**) Histologic features of carcinoma showing thymus-like differentiation (hematoxylin and eosin staining). (**B**) Extensive areas of fibrosis with groups of epithelioid cells, medium and large vesicular nuclei with small, eosinophilic nucleoli, reduced eosinophilic cytoplasm, peritumoral lymphocystic infiltration.

## Data Availability

Not applicable.

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
