# Peer review of "Thyroid Carcinoma Showing Thymus-like Differentiation (CASTLE): A Case Report"

_life, 2022, doi:10.3390/life12091314_

Round 1
Reviewer 1 Report
Dear Authors,
I read with interest this manuscript submitted for publication in Life.
Here are my comments and suggestions.
1. Title should be modified as following: Thyroid carcinoma showing thymus-like differentiation (CASTLE):.........
2. I found several ripetitive phrases. I suggest to revise the whole text avoiding these. In addition, moderate English changes are required.
3. A case-based review will improve scientific soundness and interest to readers. Please, consider this my suggestion and modify Your manuscript accordingly.
4. In general, references should be expanded. In particular, Your sentence in lines 117 and 118 must have more bibliographical references.
Minor Comment:
Miyauchi's report was in 1985 and not in 1981. Do You agree ?
2. Case report (and not "2. Materials and Methods").
Please, consider a clearer figure 1 b.
Conclusions: what You wrote in lines 174-176 is not necessary because obvious. I would take these sentences off.
Author Response
Dear Reviewer,
Thank you very much for your review and constructive suggestions.
Q1. Title should be modified as following: Thyroid carcinoma showing thymus-like differentiation (CASTLE):.........
A1: Thanks for the suggestion. We have modified the title according to the recommendations:” Thyroid carcinoma showing thymus-like differentiation (CASTLE):A case report”
Q2: I found several ripetitive phrases. I suggest to revise the whole text avoiding these. In addition, moderate English changes are required.
A2: We have made extensive editing of the paper (both the information and the page display of the text) to improve its quality and to make it easier to follow. We have re-written some paragraphs in order to convey the information more clearly and we have also eliminated repeating information.
Q3: A case-based review will improve scientific soundness and interest to readers. Please, consider this my suggestion and modify Your manuscript accordingly.
A3: In the manuscript proposed by us for publication in Life magazine, we intended to emphasize the importance of correct diagnosis in rare pathologies, in this case CASTLE disease. A review with cases already existing in the literature related to our case has been made in the Discussion section. A more extensive analysis of the cases would totally change our article and this is not the purpose, we proposed it for publication as a Case Report not as a Review. Thank you for your understanding.
Q4:In general, references should be expanded. In particular, Your sentence in lines 117 and 118 must have more bibliographical references.
A4: We totally agree with you. We have added several references both to that statement and to the whole manuscript.
According to the relevant literature, CASTLE has a mean age at diagnosis of 50-year-old, with a female-to-male ratio of 1.3:1 [6,7,8,11].
Q5: Miyauchi's report was in 1985 and not in 1981. Do You agree ?
A5: We apologize for our negligence. We have changed the year accordingly. Thank you for pointing this out.
In 2004, the World Health Organization (WHO) classified CASTLE as a both clinically and pathologically independent thyroid tumor entity, after being first described as an „intrathyroidal epithelial thymoma” in 1985 by Miyauchi et al [9,10].
Q6: Case report (and not "2. Materials and Methods").
A6: Thank you for your suggestion. We changed the subtitle “material and methods” into “Case report”.
Q7: Please, consider a clearer figure 1 b.
A7: Unfortunately, that is the only picture that we have. We tried to enhance the quality.
Q8: Conclusions: what You wrote in lines 174-176 is not necessary because obvious. I would take these sentences off.
A8: We have modified it as recommended. We thank you again for your advice.
Thank you very much for your consideration regarding this paper. We look forward to your reply.

Reviewer 2 Report
In the manuscript " The CASTLE tumor: the importance of accurate diagnosis of a rare thyroid malignancy. A cese report ", the authors present a case report of a patient diagnosed with a rare thyroid tumor. However, major criticisms are present, as follows:
- In the introduction there is no prevalence or incidence mentioned, are they more prevalent in adults. Are there present in children?
- Please discuss about the pathophysiology of the tumor
- In the Material and method part – the authors present the patient as being misdiagnosed in another Endocrinology unit, please note this in a nicer manner.
- There are no echographic images with the actual tumor. Please add them.
- What do the current guidelines stand for CD5 staining, when should we demand it?
- Why do you consider that were the favorable factors for the good evolution of the patient?
- Do we consider TSH suppression in this case?
Author Response
Dear Reviewer,
Thank you very much for your review and constructive suggestions.
We have made extensive editing of the paper (both the information and the page display of the text) to improve its quality and to make it easier to follow. We have re-written some paragraphs in order to convey the information more clearly and we have also eliminated repeating information.
Q1: In the introduction there is no prevalence or incidence mentioned, are they more prevalent in adults. Are there present in children?
A1: We have added both prevalence and incidence data in the introduction. So far, no cases of CASTLE have been reported among children.
The incidence of CASTLE among thyroid malignancies is estimated to be less than 0.1% [4,5]. Both genders are similarly affected, with a slight female predominance, with an F: M ratio of 1.3:1 [6,7]. It is most common during the fourth and fifth decades of life, with the average age at onset occurring around 48, 5 years [6,7,8]. Spindle-shaped epithelial tumor with thymus-like differentiation (SETTLE) is an unusual thyroid tumor common in the pediatric and young adult age group [12 ]. It predominantly occurs in young people especially children, adolescents, and young adults but may be found in the middle age as it has been seen in patients aged 4‐59 years [13].
Q2: Please discuss about the pathophysiology of the tumor.
A2: Thanks for the suggestion. I have added data on the pathophysiology of the tumor.
Histologically, CASTLE is very similar to thymic tissue. It is thought to arise either from ectopic thymus tissue or from the remains related to thymic development in or adjacent to the thyroid. This hypothesis is based on the fact that the tumor usually arises at the lower pole of the thyroid and exhibits several features of thymic differentiation, such as: lobulation on cut surfaces; an expansive growth pattern; thick, fibrous bands dividing tumor cell nests; many lymphocytes; perivascular spaces with lymphocytes; sparse or rare mitoses; and oval, vesicular nuclei, well-defined nucleoli, and pale cytoplasm. This tumor is also lacking foci of papillary, follicular, medullary or anaplastic carcinoma [11].
Q3: In the Material and method part – the authors present the patient as being misdiagnosed in another Endocrinology unit, please note this in a nicer manner.
A3: We apologize for the expression error, we have modified it according to your requirements.
After being initially diagnosed with undifferentiated thyroid carcinoma in another Endocrinology Service, a 50-year-old female patient was referred to our Endocrinology Clinic for further investigation of an 18-month history of persistent dyspnea, dry cough, loss of weight, lack of appetite, and fatigue, associated with dysphagia and dysphonia.
Q4: There are no echographic images with the actual tumor. Please add them.
A4: Thanks for the suggestion with the ultrasound image. We also considered adding it, but decided not to attach it to the manuscript due to its poor quality. However, with your kind encouragement, we have added the ultrasound image.
Figure 1. Cervical ultrasonography showing a hypoechogenic, finely inhomogeneous right thyroid lobe and the left thyroid lobe entirely occupied by a nodular lesion, with a hypoechoic and inhomogeneous echotexture.
Q5:What do the current guidelines stand for CD5 staining, when should we demand it?
A5: CD5 is a protein that is normally produced by specialized immune cells called T cells. Most lymphomas that start from T cells, including peripheral T-cell lymphoma, anaplastic large cell lymphoma, and extranodal T-cell lymphoma, produce CD5. Abnormal B cells can also make CD5, and some lymphomas that start from these B cells, such as chronic lymphocytic leukemia (CLL) and mantle cell lymphoma, also make CD5.
CD5 is a transmembrane protein associated with the T cell receptor that is expressed by all mature T cells and some leukemic B cells. It negatively modulates T cell activation and differentiation, and is also expressed in thymic carcinoma [10]. Therefore, CD5 is used as a marker of thymic origin. Ito et al. [6] reported a sensitivity and specificity of 82 and 100%, respectively, for CD5-based diagnoses of CASTLE. Although its negative expression does not completely rule out CASTLE, CD5 can sufficiently help us diagnose CASTLE.
An experienced histologist requests the performance of CD5 when the histopathological examination does not clearly diagnose differentiated cancer (papillary, follicular) or medullary cancer and when the usual immunohistochemistry panel shows negative results for marking with calcitonin, P63, synaptophysin, Thyroglobulin and chromogranin.
In most cases, the immunohistochemical test shows CD5-positive staining and negative staining with thyroid gland markers such as thyroglobulin and calcitonin. Molecular analysis reveals that the majority of thymic tumors are p63-positive, whereas cystic carcinoma and poorly differentiated thyroid forms are negative [3]. Therefore, CD5 is used as a marker of thymic origin. Ito et al. [11] reported a sensitivity and specificity of 82 and 100%, respectively, for CD5-based diagnoses of CASTLE. Although its negative expression does not completely rule out CASTLE, CD5 can sufficiently help us diagnose CASTLE.
An experienced histologist requests the performance of CD5 when the histopathological examination does not clearly diagnose differentiated cancer (papillary, follicular) or medullary cancer and when the usual immunohistochemistry panel shows negative results for marking with calcitonin, P63, synaptophysin, Thyroglobulin and chromogranin.
Q6: Why do you consider that were the favorable factors for the good evolution of the patient?
A6:We consider that the radiotherapy performed preoperatively and the experience of the surgeon who used a radical tumor removal technique with deep neck lymphadenectomy led to a favorable evolution of the case.
Q7:Do we consider TSH suppression in this case?
A7: No! In no case of CASTLE the TSH suppression treatment is not recommended because this type of therapy is necessary only in cases of thyroid cancers that derive from the thyroid follicle (TSH-dependent cancers) like papilary and folicular thyroid cancer. In CASTLE, as in undifferentiated (anaplastic) thyroid cancer, TSH suppression treatment has no physiopathological support.
Thank you very much for your consideration regarding this paper. We look forward to your reply.

Round 2
Reviewer 1 Report
Dear Authors,
all my comments and suggestions were satisfactorily met in the revised version of your manuscript.
Quality of presentation and scientific soundness improved, and your manuscript should be accepted for publication in Life.
Reviewer 2 Report
The changes are concordant with the report. The paper is well written and brigs value to the literature. Accept in current form.